# Community-based health insurance dropout and its determinants among women in Sidama National Regional State, Southern Ethiopia, 2024: A multilevel analysis

Kare Chawicha Debessa[1]*, Keneni Gutema Negeri[1], Mesay Hailu Dangisso[2]

**1** School of Public Health, College of Medicine and Health Sciences, Hawassa University, Hawassa, Ethiopia, **2** Ethiopian Public Health Institute, Addis Ababa, Ethiopia

* kare.debessa@gmail.com

## Abstract

### Introduction

Ensuring accessible and affordable healthcare continues to be a key priority in low- and middle-income countries, including Ethiopia. To enhance healthcare coverage and provide financial protection, especially for vulnerable populations such as women, community-based health insurance (CBHI) schemes have been introduced. Despite these efforts, dropout rates from the program remain a challenge, threatening its long-term sustainability. This study investigated determinants of CBHI dropout among women in Sidama Regional State, Ethiopia.

### Methods

A community-based cross-sectional study was conducted between December 15, 2023, and January 12, 2024, involving 835 women aged 18 years and above in the central zone of Sidama. Participants were selected via a multistage sampling technique. Data were collected through structured, pre-tested face-to-face interviews encompassing demographic, household, and scheme-related factors. Multilevel logistic regression analysis was performed using Stata version 17 to identify factors associated with CBHI dropout. Adjusted Odds Ratios (AOR) with 95% Confidence Intervals (CI) were reported, with statistical significance set at p < 0.05.

### Findings

Among the 845 women sampled, 835 participated in the interviews with a response rate of 98.8%. Of these, 77 women (9%) dropped from the CBHI scheme (AOR = 9; 95% CI: 7.43% −11.38%; p < 0.001). Lower dropout likelihood was significantly associated with increased age (AOR = 0.93; 95% CI: 0.89–0.97; p < 0.001) and larger family size (AOR = 0.28; 95% CI: 0.17–0.50; p < 0.001).

**Data availability statement:** All relevant data are within the manuscript and its supporting information files.

**Funding:** This study was financially supported by the Sidama National Regional Health Bureau.The funding organization had no role in the conceptualization, design, data analysis, manuscript preparation, and publication. There was no additional external funding received for this study.

**Competing interests:** The authors have declared that no competing interests exist.

Conversely, reduced transparency in decision-making (AOR = 2.05; 95% CI: 1.39–3.00; p < 0.001) and insufficient advocacy (AOR = 3.01; 95% CI: 2.29–7.00; p = 0.011) were positively associated with dropout. At the community level, residing in low-poverty areas (AOR = 0.29; 95% CI: 0.09–0.98; p = 0.046) and high-autonomy communities (AOR = 0.14; 95% CI: 0.04–0.50; p = 0.002) were associated with reduced dropout rates.

## Conclusion

This study identified factors linked with women's dropout from the CBHI program that threaten sustainability. Lower dropout rates were linked with older age and larger family size, while poor transparency in decision-making and weak advocacy were associated with increased dropout rates. Communities residing in low-poverty and high-autonomy areas were also associated with decreased dropouts.

## Introduction

Community-based health insurance (CBHI) has emerged as a pivotal strategy for enhancing healthcare accessibility and ensuring financial protection, particularly in low- and middle-income countries [1]. With escalating healthcare service costs and the persistent challenge of poverty, CBHI schemes aim to mitigate the financial risks associated with health-related expenditures by pooling resources from their members [2].

This collective approach not only fosters mutual support but also empowers communities to take charge of their health needs, ultimately promoting better health outcomes [3]. The significance of CBHI is underscored by its potential to address inherent inequalities in healthcare access, especially among marginalized populations like women.

Globally, financial protection mechanisms, such as CBHI, are recognized as critical components for achieving universal health coverage (UHC) [4]. In Ethiopia, where the healthcare system faces numerous challenges, including high disease burdens, inadequate health infrastructure, and a predominantly rural population [5] CBHI initiatives have gained momentum. These initiatives have proven instrumental in broadening health insurance coverage and improving the health status of communities, particularly of women [6].

Women play a crucial role as decision-makers regarding health choices within households; however, they often encounter challenges in accessing and maintaining community-based health insurance, particularly in low-resource settings [7].

High dropout rates among households enrolled in CBHI schemes are also a pressing concern that undermines the overall effectiveness of these programs [8]. Therefore, it becomes imperative to investigate the multifaceted factors that influence participation and retention among women in CBHI schemes, recognizing both individual and community dynamics that shape these experiences.

Despite the concerted efforts to implement CBHI programs in southern Ethiopia, especially in the Sidama National Regional State, there remains a concerning trend of high dropout rates (50%) among household participants [9].

This trend is particularly troubling as it directly hampers the overarching goals of CBHI to ensure universal health coverage and financial protection. Households that drop out of this program grapple with barriers, including socioeconomic challenges, cultural norms, and inadequate information regarding available benefits [10–12].

The implications of high dropout rates extend beyond the health outcomes of households themselves; this also affects their families and communities [13–15]. This underscored the critical need for a deeper understanding of the determinants contributing to these CBHI dropout rates.

The theoretical discourse surrounding community-based health insurance draws on various models, such as social capital and collective efficacy theories. Social capital theory posits that community networks, norms, and trust facilitate cooperation among members, allowing them to work collectively toward common goals [16].

Within the context of CBHI, high levels of social capital within a community can enhance trust in the community-based health insurance scheme, thereby increasing enrollment rates and retention among members [17–20].

This notion extends to women, wherein strong networks of support and information sharing can empower them to make informed health decisions and advocate for their needs within the community-based health insurance system [21].

In parallel, the collective efficacy model emphasizes the capacity of communities to mobilize resources and act collectively to achieve desired health outcomes [18]. High collective efficacy within a community correlates with a greater likelihood of engaging in health-promoting behaviors and advocating effectively for health-related changes [22].

This theoretical framework underscores the potential for community-driven initiatives, such as CBHI, to resonate with women's health needs, provided that these initiatives are grounded within structures that promote social cohesion and collective action.

An empirical study examining determinants of community-based health insurance enrollment reveals a complex interplay of socioeconomic, cultural, and informational factors that influence engagement in CBHI schemes [20,23,24].

A substantial body of studies highlighted the significance of economic determinants in shaping the likelihood of both CBHI enrollment and retention [23,25]. For instance, studies have shown that lower-income households exhibit higher dropout rates due to financial constraints, thereby emphasizing the impact of economic stability on community-based health insurance sustainability [8].

Moreover, the gendered nature of these dynamics is especially pronounced, as women often face heightened vulnerabilities stemming from socio-cultural barriers that restrict their access to information and decision-making powers related to community-based health insurance [26,27].

In addition, inadequate communication about the benefits of CBHI programs has resulted in persistent gender disparities in retention rates, thereby underscoring the necessity for targeted outreach and education initiatives that cater specifically to women's unique needs and concerns [10,28,29].

In the East African context, research has documented the vital role of community engagement and cultural perceptions in shaping health-seeking behaviors [30]. Systematic reviews emphasized that interventions aimed at increasing awareness and understanding of CBHI systems are critical for improving enrollment rates among women [31].

Specifically, cultural norms often dictated women's health-seeking behaviors and decision-making capacities, directly influencing their enrollment in the community-based health insurance program [32,33]. Studies have revealed that women with higher levels of education are better positioned to navigate the complexities of community-based health insurance, subsequently impacting their decisions to enroll and their retention within CBHI schemes [34,35].

Existing literature on community-based health insurance (CBHI) program dropout provides useful insights but leaves important gaps about women's experiences, especially in local areas like the Sidama National Regional State. Most studies focus on broad regional data and individual factors. These often miss the complex social and cultural influences,

including gender, socio-economic status, and cultural background, that affect women's enrollment and dropout from CBHI schemes [36].

This gap limits understanding of what drives women's dropout from the program. This study utilized multilevel analysis to examine both individual and community-level factors that influence dropout rates among women in Sidama. The findings aim to help policymakers and program managers create strategies to improve women's enrollment and retention in CBHI, supporting better health outcomes and program success at the local level.

## Methods

### Study area

The study was conducted in the Central Zone of the Sidama Region, Ethiopia. This region became a new regional state in June 2020 [37]. The Central Sidama Zone has six districts and one town administration. Its total population is 956,967 [38]. The study focused on Dale Woreda and Yirgalem City Administration. The study area is located about 45 km south of the regional capital, Hawassa [39].

These sites include both urban and rural areas. They show the diverse population and healthcare needs of the zone (Molla 2024). Households were randomly selected from these areas. The study assessed their enrollment in community-based health insurance (CBHI) and identified key factors affecting it. The findings help inform health policy and programs in the Sidama National Regional State [40].

### Study design and period

A community-based cross-sectional study design was conducted between December 15, 2023, and January 12, 2024. This study was based on the STROBE checklist, and it is provided as S1 File.

### Source and study population

The source population comprised women residing in households within Dale Woreda and Yirgalem City Administration in the Central Sidama Zone. The study population included women aged 18 years and older from these households.

### Inclusion and exclusion criteria

The study included women aged 18 years or older who had resided continuously in Dale Woreda and Yirgalem City Administration for at least one year were eligible for inclusion, ensuring adequate exposure to local conditions.

Exclusion criteria comprised inability to communicate due to illness, cognitive impairment, temporary absence during data collection, incapacity to provide informed consent, and new visitors likely to bias participation. These criteria were established to maintain methodological rigor and ethical standards per STROBE guidelines.

### Sampling procedure and sample size

The sample size for this study was determined using OpenEpi 3 software, referencing findings from a previous study based on a proportion of 49% from the Tigray region, Ethiopia [41,42]. The calculation was based on a 5% margin of error, and a 95% confidence level (Z = 1.96). The formula applied was:

$$n = \frac{Z^2 \cdot p \cdot (1-p)}{E^2}$$

Where $n$ is the required sample size, $Z$ is the standard normal value at the desired confidence level, $p$ is the estimated prevalence, and $E$ is the margin of error. Substituting the values ($Z = 1.96$, $p = 0.49$, and $E = 0.05$), the calculation yielded:

$$n = \frac{(1.96)^2 \cdot 0.49 \cdot 0.51}{(0.05)^2} \approx 384$$

To account for the clustering effect inherent in the sampling design, the sample size was multiplied by a design effect (DEFF) of 2, as recommended for community-based cluster surveys [43]. This adjustment resulted in an initial sample size of 768, which was rounded to 768 for practical purposes.

The determination of the minimum number of clusters required was guided by the intraclass correlation coefficient (ICC), which measures the degree of similarity within clusters. Since previous studies did not report an ICC value, a typical value of 0.01 was adopted, consistent with the range of 0.01 to 0.05 [44] for health research.

The minimum number of clusters was thus calculated as the product of the sample size and the ICC, resulting in approximately eight clusters ($768 \times 0.01 = 7.68$). However, to enhance the statistical power and representativeness of the study, the number of clusters was increased to 14. In this context, kebeles (the smallest administrative units in Ethiopia) were considered as clusters.

The average cluster size was calculated by dividing the total sample size by the number of clusters, yielding approximately 54.93 participants per cluster ($768/14$). The design effect was then recalculated using the formula $DEFF = 1 + (n-1) \times ICC$, where *n* represents the average cluster size. This calculation produced a design effect of 1.54. Nevertheless, to further increase the robustness and power of the study, the design effect was conservatively set at 2.

Finally, to adjust for a potential non-response rate of 10%, the sample size was inflated accordingly, resulting in a final estimated sample size of 845. This systematic approach to sample size determination, incorporating both statistical theory and empirical recommendations, ensures that the study is adequately powered and methodologically sound [45].

The study utilized a multistage sampling technique to select participants. This approach ensured representativeness, methodological rigor, and feasibility. First, Yirgalem City and Dale Woreda were purposively selected based on criteria relevant to the study objectives (Fig 1). Then, within each Woreda, kebeles, the smallest administrative units, were randomly selected as clusters.

In total, fourteen *kebeles* were randomly selected: nine from Dale Woreda (eight rural and one urban) and five from Yirgalem City Administration (three rural and two urban). The use of simple random sampling minimized selection bias and allowed for the accurate identification of *kebeles* using geographic maps.

Next, the total sample size was proportionally allocated to each *kebele* according to its population size. Therefore, this proportional allocation improved the representativeness of the sample across different population groups. Although clustering was applied at the *kebele* level, the number of eligible women per *kebele* varied, ranging from approximately 873–2,192.

Thus, a full cluster sampling approach was impractical. Instead, simple random sampling was used to select eligible women from lists compiled through community health records. This approach ensured that the required sample size was met within each kebele without compromising statistical validity.

At the household level, proportional allocation and simple random sampling were used to select households within the chosen *kebeles*. Subsequently, data collectors approached these households to identify eligible women. Women were prioritized because of their important roles in health-related issues at the household level.

In households where more than one eligible woman resided, we used the lottery method to select one participant randomly. Each eligible woman was assigned a unique number, which was written on a separate piece of paper. These were placed in a container, and one number was drawn by the interviewer to determine the selected respondent. This procedure ensured an unbiased and fair selection of participants within households, minimizing selection bias.

Moreover, data collectors made up to three contact attempts for households that were initially unavailable. They carefully recorded details such as household ID, enumerator ID, and the date and time of each contact attempt on tracking sheets.

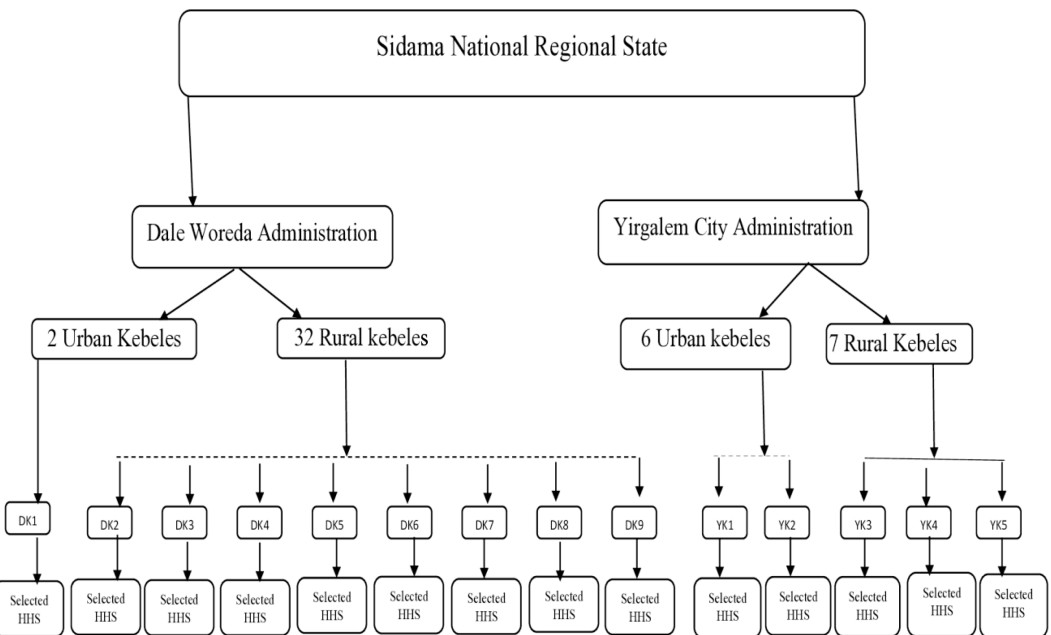

**Fig 1. Schematic diagram illustrating the multistage sampling technique used to select households for the study in Dale Woreda and Yirgalem City Administrations, Sidama National Regional State.** HHs means selected households for the interview. On the other hand, DK1, DK2, DK3, DK4, DK5, DK6, DK7, DK8, and DK9 represent the nine *kebeles* randomly selected from Dale *woreda*, while YK1, YK2, YK3, YK4, and YK5 represent the five *kebeles* randomly selected from Yirgalem city administration. Households were selected proportionally from each *kebele* based on their population size.

## Study variables

The dependent variable was the CBHI dropout rate, assessed through self-reported information from women (binary: yes or no). Independent variables were categorized as individual-level (socioeconomic, demographic, and CBHI-related factors) and community-level (residence, community-level women's literacy, community-level women's poverty, and community-level women's autonomy). The details of variable measurement were provided in S2 File.

## Data collection and quality assurance

A structured questionnaire (S3 File), adapted from previous studies [46,47], was employed for data collection. Validated survey tools were integrated to enhance reliability; initial drafts were translated into *Sidaamu Afoo*, followed by back-translation to ensure coherence. A pretest with a 5% sample in *Hitata kebele*, *Tabor* sub-city, Hawassa, provided feedback for refining the tool's wording, coding errors, and cultural sensitivity, ensuring ease of administration during interviews.

The reliability of data collection tools was assessed using Cronbach's alpha, achieving a score of 0.9. Twenty-five data collectors with bachelor's degrees in health sciences conducted face-to-face interviews via the ODK mobile application, supervised by five experienced public health supervisors who were master's holders.

Quality control measures implemented by the study team included training, pre-testing of tools, reviews, and daily checks to ensure data accuracy and validity, completeness, and adherence to ethical standards.

## Data analysis

Before analysis, variables (S4 File) were recorded and categorized. Categorical variables were expressed as absolute frequencies and percentages, while numerical variables were summarized using means and standard deviations. The wealth index was derived from PCA based on ownership of various household assets.

A multilevel logistic regression model was employed to estimate adjusted odds ratios (AORs) with 95% confidence intervals (CIs) for outcome determinants. To validate the necessity of a multilevel model, a random intercept model of a multilevel logistic regression was applied, generating intraclass correlation coefficients (ICCs).

A model was warranted if the ICC exceeded 5%. variables with p-values < 0.25 from bivariable analysis, alongside other literature-supported variables, were included in the multivariate analysis model. Additionally, interaction terms were assessed for effect modification, and multicollinearity was evaluated using variance inflation factor thresholds (<5).

To account for the data's hierarchical structure, four models were analyzed: Model 0 (empty model), Model 1 (individual-level variables), Model 2 (community-level variables), and Model 3 (combined individual- and community-level variables).

This multi-stage sampling design not only facilitated participant selection but also strengthened the reliability and validity of the data collected. Statistically significant associations were determined using AORs with 95% Cs and P values <0.05.

### Ethics statement

This study adhered to ethical standards as outlined in the Declaration of Helsinki. Ethical approval for the research was granted by the Institutional Review Board (IRB) (S5 File) of the College of Medicine and Health Sciences at Hawassa University under reference number IRB/021/16. Letters of support were obtained from Hawassa University School of Public Health, the Sidama Region Health Bureau, district health offices, and kebele administrators.

Informed written consent was obtained from all study participants meeting the inclusion criteria before interviews. Participants were briefed on the study's objectives, data collection methods, voluntary nature, confidentiality, and potential risks and benefits.

Confidentiality was maintained throughout data collection and storage. Participation posed a minimal risk, limited to approximately forty minutes for the interview; participants had the right to decline to answer any uncomfortable questions.

### Results

The study focused on women from the central zone of the Sidama region, Ethiopia, assessing their sociodemographic, economic, and CBHI-related characteristics. Of the 845 women sampled, 835 women completed the interviews, yielding a response rate of 98.8%. Among these participants, 77 (9%, 95% CI: 7.43–11.38; p < 0.001) had withdrawn from the CBHI scheme, whereas 758 (91%) remained enrolled (Fig 2).

Among the study participants, 484 (58%) resided in rural areas, while 351 (42%) were urban residents. Community-level women's literacy was a significant concern: 441 (52.8%) of women were classified as illiterate. Regarding community-level women's autonomy, 431 (51.6%) were considered autonomous, compared to 404 (48.4%) who were not (Table 1).

In terms of CBHI scheme features, 258 (30.9%) respondents reported that the benefit packages were fully relevant, whereas 577 (69.1%) perceived the packages as only partly included. Transparency in CBHI decision-making was also a concern: 270 (32.3%) women viewed decisions as extremely transparent, while 565 (67.7%) considered them only somewhat transparent (Table 1).

The adequacy of CBHI promotion was nearly evenly divided: 420 (50.3%) women rated it as adequate, and 415 (49.7%) as inadequate. Acceptance of CBHI as a health financing strategy was low, with 304 (36.4%) women regarding it as acceptable, compared to 539 (64.6%) who did not (Table 1).

The mean age of participants was 38.72 years (SD = 14.105), reflecting a predominantly working-age group, and the average family size was 4.91 members (SD = 1.346), consistent with small to medium-sized households. Healthcare service utilization was low, with an average of 0.85 visits (SD = 1.259). Satisfaction with health services was highly variable, with a mean score of 12.59 (SD = 17.76). For additional sociodemographic and CBHI-related characteristics, please refer to Table 1.

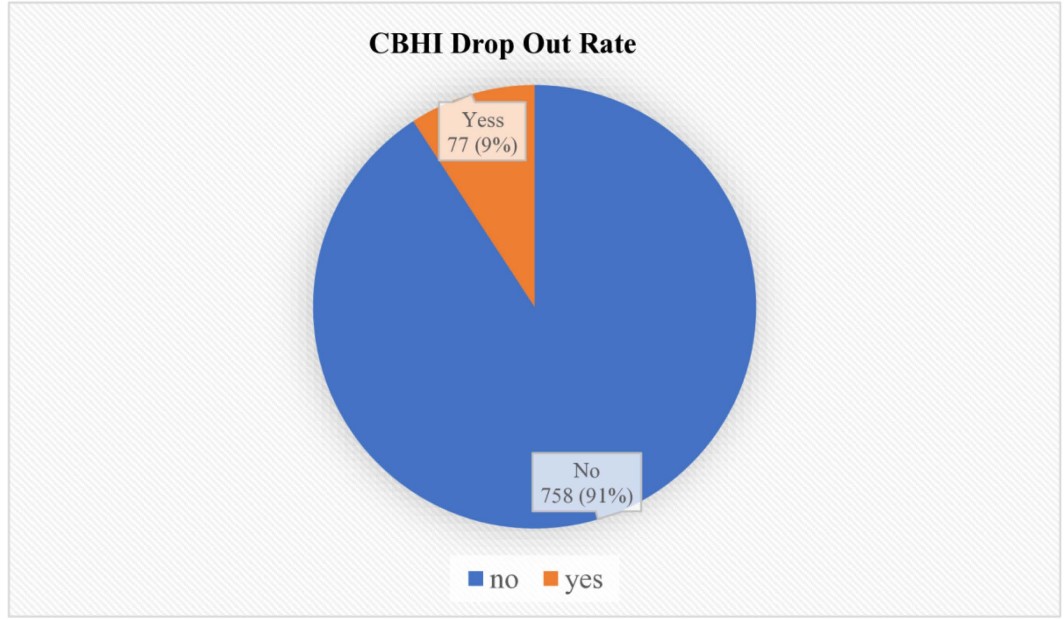

**Fig 2. Community-based health insurance dropout rate among women in Sidama National Regional State, Ethiopia, 2024 (n = 835).**

## Random effects model information

The results from the multilevel logistic regression analysis revealed important insights into the factors influencing community-based health insurance (CBHI) membership dropout rates, particularly through the intraclass correlation coefficient (ICC).

The ICC for the kebele code level was calculated at 0.351, with a standard error of 0.1022 and a 95% confidence interval ranging from 0.1836 to 0.5661 (Table 2). This indicated that approximately 35.14% of the total variance in CBHI membership dropout rates is attributable to differences between *kebele* (community) levels.

The moderate ICC value suggested that women within the same kebele exhibited more similar dropout rates compared to those from different kebeles. This finding highlighted the significance of community-level factors in influencing dropout rates, indicating that interventions should be tailored to address community-specific contexts rather than focusing solely on individual characteristics.

In summary, the ICC results demonstrated that a substantial portion of the variance in CBHI membership dropout rates is linked to *kebele*-level differences, emphasizing the need for multilevel analysis to effectively understand and address dropout dynamics within community-based health insurance programs.

## Model selection

The details of the model selection description were provided in S6 File.

**Multivariate analysis results of community-based health insurance dropout and its determinants among women in Sidama National Regional State, Ethiopia, 2024 (n = 835).**

The multilevel logistic regression analysis results identified several individual and community-level significant variables associated with CBHI membership dropout among women.

First, women residing in communities with low-level poverty exhibited significantly lower odds of dropout compared to those in higher-poverty levels, with an adjusted odds ratio (AOR) of 0.29 (95% CI: 0.09 to 0.98; p = 0.046). This finding

**Table 1. Sociodemographic, economic, and CBHI scheme-related characteristics of the women in the central zone of Sidama region, Ethiopia, 2024 (n=835).**

| Variables | Categories | Frequency | % |
|---|---|---|---|
| Residence | Urban | 351 | 42.0 |
| | Rural | 484 | 58.0 |
| Administrative level | Dale woreda administration | 537 | 64.3 |
| | Yirgalem city administration | 298 | 35.9 |
| Recorded religion | Protestant | 737 | 88.3 |
| | Others | 98 | 11.7 |
| Recoded ethnic group | Sidama | 797 | 95.5 |
| | Others | 38 | 4.6 |
| Wealth index | Lowest | 132 | 15.8 |
| | Second | 130 | 15.6 |
| | Middle | 223 | 26.7 |
| | Fourth | 186 | 22.3 |
| | Highest | 164 | 19.6 |
| Community-level women's literacy | Illiterate | 441 | 52.8 |
| | Literate | 394 | 47.2 |
| Community-level women's autonomy | Low | 404 | 48.4 |
| | High | 431 | 51.6 |
| Recoded education | Not attended formal education | 367 | 44.0 |
| | Attended education | 468 | 56.0 |
| Women autonomy | Un autonomous | 373 | 44.7 |
| | Autonomous | 462 | 55.3 |
| Recoded_marital_status | Married | 780 | 93.3 |
| | Others | 55 | 6.6 |
| Community-level women's poverty | High | 468 | 56.0 |
| | Low | 367 | 44.0 |
| Household head reported by women | Male | 741 | 88.7 |
| | Female | 94 | 11.3 |
| CBHI membership | No | 544 | 65.1 |
| | Yes | 291 | 34.9 |
| Benefit packages | Relevant packages included | 258 | 30.9 |
| | Partly included | 577 | 69.1 |
| CBHI premium fair | Fair | 385 | 46.1 |
| | It is not fair | 450 | 53.9 |
| CBHI decisions transparent | Extremely transparent | 270 | 32.3 |
| | Somewhat transparent | 565 | 67.7 |
| CBHI decisions inclusive | Extremely inclusive | 207 | 24.8 |
| | Partly inclusive | 628 | 75.2 |
| CBHI promotion adequate | Adequate | 420 | 50.3 |
| | Not adequate | 415 | 49.7 |
| CBHI acceptable strategy | No | 539 | 64.6 |
| | Yes | 304 | 36.4 |

**Table 2. The intraclass correlation coefficient of the community-based health insurance dropouts among women in the central zone of Sidama region, Ethiopia, 2024 (n = 835).**

| Level | ICC | Standard error | (95% confidence interval) |
|---|---|---|---|
| Kebele | 0.35 | 0.102 | 0.18, 0.57 |

indicated that living in a community with low-level poverty was associated with a 71% reduction in the odds of dropout (Table 3).

Similarly, women from autonomous communities demonstrated significantly lower odds of dropout than those from less autonomous communities (AOR = 0.14, 95% CI: 0.04 to 0.49; p = 0.002). This suggested that higher levels of community autonomy were linked to an 86% decrease in the odds of dropout (Table 3).

Additionally, age emerged as a significant factor; each additional year was associated with a decrease in the odds of dropout (AOR = 0.93, 95% CI: 0.89 to 0.97; p < 0.001). This finding indicated that for every year increase in age, the odds of dropout decreased by approximately 7% (Table 3).

In addition, larger family sizes were associated with significantly lower odds of dropout compared to smaller families (AOR = 0.28, 95% CI: 0.17 to 0.50; p < 0.001). This underscored the importance of family dynamics, as women from larger families had about a 72% lower likelihood of dropping out (Table 3).

Conversely, transparency in decision-making related to CBHI was found to be critical; women in communities where decision-making lacked transparency had significantly higher odds of dropout compared to those in transparent environments (AOR = 2.05, 95% CI: 1.39 to 3.00; p < 0.001) (Table 3). This finding suggested that insufficient transparency resulted in approximately twice the odds of dropout among women.

Additionally, women reporting inadequate CBHI advocacy efforts faced significantly higher odds of dropout compared to those who experienced adequate advocacy (AOR = 3.01, 95% CI: 1.29 to 7.00; p = 0.011). This indicated that inadequate advocacy was associated with nearly a threefold increase in the odds of dropout (Table 3).

Overall, these findings underscore the multifaceted factors influencing CBHI membership retention and emphasize the need for targeted interventions that address socioeconomic and community dynamics to improve engagement in health insurance programs among women in Ethiopia.

## Discussion

The study conducted in the central zone of Sidama National Regional State, Ethiopia, revealed a dropout rate of 9% (95% CI: 7.43%−11.38%), with 77 women discontinuing their CBHI membership and 758 (91%) women remaining enrolled.

Dropout rates in CBHI programs demonstrated significant variations, indicating considerable retention challenges across different contexts. Comparative analyses indicate that dropout rates frequently exceed the 9.22% observed in this study; for instance, Arba Minch in Southern Ethiopia reported a dropout rate of 21.5% [48], while Uganda reported a rate of 25.1% [49]. Other notable rates include 31.9% in Jimma, Ethiopia [50], and 32.4% in the Oromia Region of Ethiopia [51].

Furthermore, dropout rates have reached as high as 37.3% in the Dera District of Ethiopia [52] and 40% in Eastern Sudan [53], in Yirgalem town, southern Ethiopia, 50% [54]. In Burkina Faso, dropout rates ranged from 30.9% to 45.7% [55], while a systematic review identified an average dropout rate of 34% in Ethiopia [8].

In contrast, Ghana experienced significant fluctuations, with dropout rates rising from 34.8% in 2012 [56] to 53% in 2016 [57]. Global reports also highlighted alarming figures, such as a dropout rate of 63% in Ahmedabad, India [58] and up to 83% in Senegal [59] underscoring the substantial challenges faced by CBHI programs.

These variations can often be attributed to socioeconomic factors such as healthcare service quality, financial constraints, and cultural attitudes toward community-based health insurance. Therefore, targeted interventions are essential

**Table 3. Determinants of CBHI dropouts among women in the central zone of Sidama region, Ethiopia, 2024 (n = 835).**

| Variables | | Have you dropped out of CBHI? | | CPR (95% CI) | APR (95% CI) | p-value |
|---|---|---|---|---|---|---|
| Age in years | | | | **0.88(0.84-0.92)** | **0.93(0.89-0.97)** | < 0.001 |
| Family size | | | | **0.28(0.19-0.40)** | **0.27(0.17-0.45)** | < 0.001 |
| Satisfaction | | | | 1.01(0.99-1.02) | 1.01(0.99-1.02) | 0.319 |
| Frequency of health facility visits | | | | 0.70(0.48-1.01) | 1.03(0.63-1.67) | 0.913 |
| Recoded marital status | Married | 704(90.3) | 76(9.7) | 1 | 1 | |
| | Others | 54(98.2) | 1(1.8) | 0.20(0.02-2.10) | 0.43(0.02-8.21) | 0.573 |
| Recoded education | Illiterate | 130(94.9) | 7(5.1) | 1 | 1 | |
| | Literate | 628(90.0) | 70(10.0) | 1.11(0.37-3.27) | 1.38(0.17-11.50) | 0.765 |
| Residence | Urban | 312(88.9) | 39(11.1) | 1 | 1 | |
| | Rural | 446(92.1) | 38(7.9) | 0.76(0.41-1.39) | 0.50(0.20-1.25) | 0.164 |
| Wealth index | Lowest | 193(91.5) | 18(8.5) | 1 | 1 | |
| | Second | 152(88.4) | 20(11.6) | 4.186(0.56-2.47) | 2.27(0.93-5.50) | 0.073 |
| | Middle | 145(90.1) | 16(9.9) | 0.76(0.38-1.53) | 1.29(0.50-3.62) | 0.629 |
| | Fourth | 132(89.8) | 15(10.2) | 0.84(0.48-1.46) | 1.16(0.43-3.11) | 0.771 |
| | Highest | 136(94.4) | 8(5.6) | 0.91(0.13-6.27) | 1.87(0.32-10.94) | 0.488 |
| Community-level women's poverty | High | 184 (79.0) | 49(21.0) | 1 | 1 | |
| | **Low** | **574(95.3)** | **28(4.7)** | **0.16(0.08-0.31)** | **0.29(0.09-0.98)** | **0.044*** |
| Community-level women's literacy | Low | 418(88.7) | 53(11.3) | 1 | 1 | |
| | High | 340(93.4) | 24(6.6) | 0.72(0.31-1.68) | 0.61(0.24-1.55) | 0.298 |
| Community-level women's autonomy | Nonautonomous | 367(84.6) | 67(15.4)) | 1 | 1 | |
| | **Autonomous** | **391(97.5)** | **10(2.5)** | **0.16(0.04-0.59)** | **0.14(0.04-0.49)** | **0.003** |
| Promotion adequate | Adequate | 478(91.2) | 46(8.8) | 1 | 1 | |
| | Not adequate | 280(90.0) | 31(10.0) | 0.85(0.42-1.73) | **3.00(1.29-7.00)** | **0.015** |
| Contribution fair | Fair | 362(94.0) | 23(6.0) | 1 | 1 | |
| | Not fair | 396(88.0) | 54(12.0) | 3.73(1.80-7.70) | 1.47(0.79-2.74) | 0.224 |
| Decision transparent | transparent | 254(94.1) | 16(5.9) | 1 | 1 | |
| | **Not transparent** | **504(89.2)** | **61(10.8)** | **2.90(1.70-4.95)** | **2.04(1.39-3.00)** | **<0.001** |
| Decision inclusive | Include all times | 197(95.2) | 10(4.8) | 1 | 1 | |
| | Include some times | 561(89.3) | 67(10.7) | 2.87(1.22-6.76) | 0.75(0.37-1.53) | 0.412 |
| Packages adequacy | Adequate | 237(91.9) | 21(8.1) | 1 | 1 | |
| | Not adequate | 521(90.3) | 56(9.7) | 2.17(1.12-4.18) | 2.08(0.74-5.87) | 0.180 |
| Acceptable strategy | No | 270(88.8) | 34(11.2) | 1 | 1 | |
| | Yes | 488(91.9) | 43(8.1) | 0.46(0.26-0.81) | 0.76(0.33-1.75) | 0.519 |

for improving women's retention in CBHI programs and necessitate strategies that enhance member engagement and satisfaction across diverse contexts [60].

The findings indicated that several individual and community-level factors were significantly associated with dropout rates among women, including community poverty levels, autonomy, age, type of residence, family size, transparency in decision-making, advocacy promotion, and literacy levels, each warranting detailed consideration.

Firstly, the study revealed that women residing in low-poverty communities were significantly less likely to drop out. Similarly, evidence from Ghana, Uganda, and Nigeria identifies economic hardship as a primary barrier to sustained CBHI enrollment [49,61,62].

This is because poor households often face difficulties affording premiums and tend to prioritize immediate survival needs over community-based health insurance contributions [63]. Moreover, irregular income streams are common among impoverished populations, complicating timely premium contributions, thereby increasing dropout risk from the CBHI schemes [64].

Furthermore, high indirect costs, such as transportation, reduce the perceived value of CBHI membership [65]. Hence, financial support mechanisms, including flexible payment options, are essential to enhance retention among economically vulnerable groups. In addition, targeted interventions such as financial subsidies for low-income women and empowerment programs have been suggested to address these challenges [66].

Additionally, community-level women's autonomy was strongly associated with lower dropout rates. Cultural norms restricting women's decision-making power, financial dependence on male household members, and limited access to information exacerbated their vulnerability to discontinuing membership [67].

Accordingly, women from communities with higher autonomy levels demonstrated higher retention, consistent with findings from Burkina Faso, where participatory governance and community involvement in CBHI decision-making improved retention [68]. This is because autonomy fosters a sense of ownership, transparency, and accountability, which are critical for building trust and commitment to the community-based health insurance scheme [6].

Conversely, CBHI programs implemented through top-down approaches without meaningful community engagement often experience distrust and higher dropout rates [69]. Empowerment initiatives that enhance women's agency have been shown to increase understanding of CBHI benefits and encourage sustained membership. Therefore, the contrast between participatory versus top-down approaches underscores the importance of community engagement and women's empowerment in improving retention [70].

Furthermore, age was inversely related to CBHI dropout. Older women were more likely to maintain membership, a pattern supported by studies from South Africa and Ethiopia demonstrating that older adults have greater health needs and risk awareness, which increases their valuation of community-based health insurance benefits [8,71].

In contrast, younger women may underestimate their health risks or face competing priorities such as childcare and employment, limiting their engagement with CBHI schemes. Consequently, this underscores the importance of targeted educational and outreach programs to address the specific barriers faced by younger women [72].

In addition, larger family size was associated with lower CBHI dropout rates. This finding is consistent with studies from Tanzania and Ethiopia, where families with more dependent members tend to maintain enrollment due to shared financial responsibilities and social support networks [8,73].

Larger households often perceive a collective benefit from CBHI coverage, which motivates continued engagement despite economic challenges. Moreover, social cohesion within larger families provides resilience against economic shocks that might otherwise drop out of the CBHI scheme. On the other hand, smaller families may lack such social support and collective financial capacity, increasing their vulnerability to CBHI dropout [74].

Moreover, transparency in CBHI decision-making emerged as a critical determinant of retention. The study found that a lack of transparency in decision-making at the district level nearly doubled the odds of dropout. This finding aligns with studies from India and other African countries showing that transparent governance builds trust and encourages continued engagement [75,76].

Transparent communication regarding financial management, benefit entitlements, and grievance redress mechanisms reduces suspicion and strengthens member confidence [77]. Therefore, CBHI programs should prioritize transparent governance practices to enhance CBHI retention. Conversely, a lack of good governance fosters distrust and disengagement, which negatively impacts membership continuity and sustainability [1].

Additionally, inadequate advocacy and promotion of the CBHI program were strongly associated with increased dropout rates. Evidence from Rwanda and other African contexts demonstrates that sustained, culturally sensitive advocacy campaigns improve enrollment and retention by raising awareness of benefits and mitigating misconceptions [78,79].

Without proper promotion of the CBHI program, beneficiaries often lack critical information, leading to disengagement and attrition from the scheme. Hence, strengthening advocacy efforts through trusted community leaders and women's groups is essential to improve retention [80].

Finally, robust community mobilization and outreach through women's groups and local leaders have proven effective in enhancing both enrollment and retention in CBHI schemes [81]. This community-driven approach contrasts with impersonal, centralized strategies and highlights the importance of culturally appropriate, grassroots engagement to sustain CBHI membership. Therefore, effective advocacy and community mobilization collectively shape CBHI dropout rates and retention patterns [82].

## Limitations of the study

This study has several notable limitations. Firstly, its cross-sectional design restricts causal inferences about the relationship between dropout determinants and community-based health insurance (CBHI) membership. The reliance on self-reported data may introduce response bias [83], thereby compromising the accuracy of reported dropout rates and influencing factors.

Another limitation of this study was the small number of dropout respondents (9.22%), which may limit the ability to identify factors specific to CBHI dropouts. The study also did not include qualitative data. This may limit the depth and context of understanding about why people dropped out.

Additionally, the study's focus is limited to the Sidama National Regional State, which may affect the generalizability of the findings to other contexts. Moreover, concentrating exclusively on women may overlook the influence of male decision-makers and broader family dynamics on CBHI participation.

While various individual and community-level factors were examined, potential unmeasured socio-cultural influences may also play a role in dropout rates. The data collection was confined to a specific timeframe, possibly missing seasonal variations in health-seeking behaviors, including CBHI enrollment. Despite a high response rate, non-response bias is a concern, as individuals who declined to participate might differ systematically from those who did.

Finally, the study's quantitative approach may have overlooked qualitative insights that could provide a more detailed understanding of participants' experiences [84] with CBHI. Acknowledging these limitations is essential for refining future research methodologies and deepening the understanding of factors contributing to CBHI dropout rates.

Therefore, the extent of these limitations might have been undervalued or overvalued, and as such, the association of these determinants with CHI dropouts might have been underestimated or overestimated.

## Conclusion

This study examined the factors influencing women's dropout from the CBHI program in the central zone of the Sidama region, Ethiopia. The results showed that women from both low-community-level poverty and high-community-level autonomy areas were less likely to leave the CBHI program. Older women and those with larger families were also more likely to remain enrolled. These findings highlight the roles of women's economic and autonomy status, age, and family size in enhancing retention in CBHI schemes.

Conversely, the study identified that poor transparency in decision-making and limited advocacy for CBHI increased dropout rates. Therefore, improving communication and transparency within the program decision-making is essential to retain members and build trust.

Based on these results, it is recommended that practitioners provide targeted education and outreach to women, especially in disadvantaged communities, to raise awareness and trust in CBHI. Policymakers should support efforts to improve economic conditions and promote women's decision-making while ensuring that CBHI processes are transparent and easy to understand. Decision makers should invest in advocacy and involve women members in CBHI planning and management, making information accessible to all.

Future research should use qualitative methods, such as interviews and focus group discussions, to better understand the reasons for CBHI dropout so as to offer deeper insights into the experiences and views of both current and former members, as well as other stakeholders.

## Supporting information

**S1 File. A STROBE check list.**
(DOCX)

**S2 File. Variables measurement.**
(DOCX)

**S3 File. Questionnaires.**
(DOCX)

**S4 File. Dataset in. CSV.**
(CSV)

**S5 File. Institutional Review Board (IRB) approval letter.**
(PDF)

**S6 File. Model fitness information.**
(DOCX)

## Acknowledgments

The authors would like to sincerely thank the study participants, data collectors, supervisors, Hawassa University College of Medicine and Health Sciences, Sidama National Regional Health Bureau, and the health offices of Dale *Woreda* and Yirgalem City for their invaluable contributions to this research. Their support and collaboration were essential to its successful completion.

## Author contributions

**Conceptualization:** Kare Chawicha Debessa, Keneni Gutema Negeri, Mesay Hailu Dangisso.

**Data curation:** Kare Chawicha Debessa, Keneni Gutema Negeri, Mesay Hailu Dangisso.

**Formal analysis:** Kare Chawicha Debessa, Keneni Gutema Negeri, Mesay Hailu Dangisso.

**Funding acquisition:** Kare Chawicha Debessa, Keneni Gutema Negeri, Mesay Hailu Dangisso.

**Investigation:** Kare Chawicha Debessa, Keneni Gutema Negeri, Mesay Hailu Dangisso.

**Methodology:** Kare Chawicha Debessa, Keneni Gutema Negeri, Mesay Hailu Dangisso.

**Project administration:** Kare Chawicha Debessa, Keneni Gutema Negeri, Mesay Hailu Dangisso.

**Resources:** Kare Chawicha Debessa.

**Software:** Kare Chawicha Debessa, Keneni Gutema Negeri, Mesay Hailu Dangisso.

**Supervision:** Kare Chawicha Debessa, Keneni Gutema Negeri, Mesay Hailu Dangisso.

**Validation:** Kare Chawicha Debessa, Keneni Gutema Negeri, Mesay Hailu Dangisso.

**Visualization:** Kare Chawicha Debessa, Keneni Gutema Negeri, Mesay Hailu Dangisso.

**Writing – original draft:** Kare Chawicha Debessa, Keneni Gutema Negeri, Mesay Hailu Dangisso.

**Writing – review & editing:** Kare Chawicha Debessa, Keneni Gutema Negeri, Mesay Hailu Dangisso.

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
