## [Decision Letter · Decision Letter 0]

30 May 2025

Dear Dr. Debessa,

Thank you for submitting your manuscript to PLOS ONE. After careful consideration, we feel that it has merit but does not fully meet PLOS ONE’s publication criteria as it currently stands. Therefore, we invite you to submit a revised version of the manuscript that addresses the points raised during the review process.

We look forward to receiving your revised manuscript.

Kind regards,

Abdene Weya Kaso, MPH

Academic Editor

PLOS ONE

Journal Requirements:

[This study was financially supported by the Sidama National Regional Health Bureau.The funding organization had no role in the conceptualization, design, data analysis, manuscript preparation, and publication.].

4. Please note that funding information should not appear in the Acknowledgments section or other areas of your manuscript. We will only publish funding information present in the Funding Statement section of the online submission. Please remove any funding-related text from the manuscript.

5. Please include captions for your Supporting Information files at the end of your manuscript, and update any in-text citations to match accordingly. Please see our Supporting Information guidelines for more information: http://journals.plos.org/plosone/s/supporting-information .

Reviewers' comments:

Reviewer's Responses to Questions

**Comments to the Author**

1. Is the manuscript technically sound, and do the data support the conclusions?

Reviewer #1: Partly

Reviewer #2: Yes

2. Has the statistical analysis been performed appropriately and rigorously?

Reviewer #1: No

Reviewer #2: Yes

3. Have the authors made all data underlying the findings in their manuscript fully available?

Reviewer #1: Yes

Reviewer #2: Yes

4. Is the manuscript presented in an intelligible fashion and written in standard English?

Reviewer #1: Yes

Reviewer #2: Yes

Reviewer #1: 1.This study is interested in dropout from CBHI and it determinants. Out of 835 respondents interviewed, only 77 (9.22%) respondents who dropped were part of this study. Remaining 91% have not dropped out, and so may not provide adequate reasons for dropout. One of the major inclusion criteria for this study could have been to include mainly those who dropped out of CBHI.

2. A study like this could have benefit more through qualitative study that will examine drop out rate and its determinants especially through those that dropped out and other stakeholders.

3. Result section of this study contained so much descriptive, knowing that such will be supported with Tables or figures.

4. Table 1 in the paper did not present the socioeconomic demographics adequately. For instance, from the sample size determination, the authors showed the region, the city administrations and urban-rural divides. These could have been presented in Table 1 to enable the reader understand the study components.

5. In lines 180-182, reliability and validity of instruments cannot be achieved together through Cronbatch's Alpha. The authors need to make a clear distinction between reliability and validity processes and their results.

6. Again the paper dwelt so much on the analysis model specifications than the actual results. For instance, Intraclass Correlation Coefficient, Model comparison and selection criteria etc are not very important for reader's comprehension of the study.

7. Table 4 and its descriptive are not aligned. While the descriptive contained the p-values, the Table itself does not contain p-values, which is very important in statistical inference.

8.Some sentences were repeated over and over from the abstract to the result section. For example, cross-sectional analysis from December 2023 to January 2024.

Reviewer #2: Dear Authors,

Your manuscript addresses a critical topic with significant relevance for national projects and policymakers. To enhance the quality and impact of your work, please consider the following suggestions:

1. Grammatical and Formatting Review: A thorough proofreading is necessary to correct grammatical errors and improve readability.

2. Typographical and Formatting Issues: There are instances of missing spaces, incorrect capitalisation, and inconsistent reference formatting.

3. Line 210: What are other factors for your exclusion?

3. Figures & Tables: Some figures (e.g., Figure 1) lack clear titles explaining what is being measured.

4. Similarity of the first paragraph of the results and the discussion.

5. Expanding the discussion on CBHI dropout rate challenges (particularly for women)

6. Your sampling technique was multistage. How did you consider the design effect?

**Do you want your identity to be public for this peer review?** For information about this choice, including consent withdrawal, please see our Privacy Policy

Reviewer #1: No

Reviewer #2: **Yes: ** Gelgelo Wodessa

---

## [Author Response · Author response to Decision Letter 1]

5 Jun 2025

Dear Dr. Kaso,

Academic Editor

PLOS ONE

Thank you for your detailed feedback and for the opportunity to revise our manuscript entitled “Community-based health insurance dropout and its determinants among women in Sidama National Regional State, Southern Ethiopia, 2024: A multilevel analysis” (Manuscript ID: PONE-D-25-06190).

We have carefully addressed all the comments and suggestions provided by the reviewers and the editorial team. In response, we have revised the manuscript and prepared the following files for resubmission:

• A response letter detailing how each reviewer and editorial comment has been addressed (“Response to Reviewers”)

• A revised manuscript with tracked changes (“Revised Manuscript with Track Changes”)

• A clean version of the revised manuscript (“Manuscript”)

All files have been uploaded according to PLOS ONE’s submission guidelines. We believe these revisions have strengthened our manuscript, and we sincerely appreciate the constructive feedback that guided these improvements.

Should you require any additional information or clarification, please do not hesitate to contact me.

Best regards,

Kare Chawicha Debessa

The corresponding author

---

## [Decision Letter · Decision Letter 1]

26 Jun 2025

Dear Dr. Debessa,

Thank you for submitting your manuscript to PLOS ONE. After careful consideration, we feel that it has merit but does not fully meet PLOS ONE’s publication criteria as it currently stands. Therefore, we invite you to submit a revised version of the manuscript that addresses the points raised during the review process.

Please submit your revised manuscript by Aug 10 2025 11:59PM. If you will need more time than this to complete your revisions, please reply to this message or contact the journal office at plosone@plos.org . A rebuttal letter that responds to each point raised by the academic editor and reviewer(s). You should upload this letter as a separate file labeled 'Response to Reviewers'.A marked-up copy of your manuscript that highlights changes made to the original version. You should upload this as a separate file labeled 'Revised Manuscript with Track Changes'.An unmarked version of your revised paper without tracked changes. You should upload this as a separate file labeled 'Manuscript'.

We look forward to receiving your revised manuscript.

Kind regards,

Abdene Weya Kaso, MPH

Academic Editor

PLOS ONE

Journal Requirements:

Reviewers' comments:

Reviewer's Responses to Questions

**Comments to the Author**

Reviewer #1: All comments have been addressed

Reviewer #2: (No Response)

2. Is the manuscript technically sound, and do the data support the conclusions?

Reviewer #1: Yes

Reviewer #2: Yes

3. Has the statistical analysis been performed appropriately and rigorously?

Reviewer #1: Yes

Reviewer #2: Yes

4. Have the authors made all data underlying the findings in their manuscript fully available?

Reviewer #1: Yes

Reviewer #2: Yes

5. Is the manuscript presented in an intelligible fashion and written in standard English?

Reviewer #1: Yes

Reviewer #2: Yes

Reviewer #1: The authors have taken time to address the issues raised in the initial review. For instance, they have included levels of statistical significance in the study. They have also included more variables of importance in the Table 1.

Reviewer #2: Dear Authors,

Thank you for your thoughtful revisions and for addressing the previous feedback. Your manuscript has undergone significant improvements in terms of structure, clarity, and depth. The additional explanations and refinements enhance the study’s quality, but I have a few further suggestions to strengthen your work.

General Comments:

• Proofread language and grammar to enhance clarity.

• Your abstract requires revision for clarity, particularly concerning the use of abbreviations and methods.

• The background needs thorough revisions; some paragraphs convey the same message. Paragraphs beginning at line numbers 144, 152, and 156 would be better merged into a single paragraph. This approach applies to all background sections to better illustrate what dropout and the factors influencing enrolment among women look like, from a global perspective down to the Sidama region context.

• There is inconsistency in the dropout rate, which is stated as 9.2% in the abstract and 9% in the figure.

• It would be more effective to describe your study area in one paragraph rather than delving into details about the selection and justification of the Woredas, which are part of the sampling procedure.

• The current sample size reported in the abstract and the document significantly differ. There are three different sample sizes mentioned within one document. Why is that? Also, what is your sampling technique? Previously, you mentioned that the multistage sampling technique was utilised. If you are to employ the clustering sampling technique within a multistage sampling technique, you might include all women in a kebele. How many women are there in one kebele? In clustering, all members of the cluster are included in the sample size.

• Line 254: If more than one woman is present in the house, how was one woman selected?

• Regarding your discussion, state your statistically significant findings, then compare, justify and recommend its impacts.

• Conclude your main finding with a recommendation to practitioners, policy and decision makers.

**Do you want your identity to be public for this peer review?** For information about this choice, including consent withdrawal, please see our Privacy Policy

Reviewer #1: **Yes: ** Eric Obikeze

Reviewer #2: **Yes: ** Gelgelo Wodessa

---

## [Author Response · Author response to Decision Letter 2]

2 Jul 2025

A point–by–point response to reviewers’ comments and journal requirements

Reviewer comments

Reviewer #1

Comments 1: All comments have been addressed.

Authors’ response: We thank the reviewer for their positive feedback and are pleased that our revisions have satisfactorily addressed all comments. We appreciate the reviewer’s time and thoughtful input, which have helped improve the quality of the manuscript.

Reviewer #2

Comment 1: Proofread language and grammar to enhance clarity.

Authors’ response: Thank you very much for this important comment, which aligns with the journal’s language and clarity requirements. We have thoroughly revised the manuscript to improve grammar, sentence structure, and overall readability. We trust that the revised version now meets the expected standard of written academic English.

Comment 2: Your abstract requires revision for clarity, particularly concerning the use of abbreviations and methods.

Authors’ response: Thank you for this helpful suggestion. We have revised the abstract to improve clarity and flow, especially in the description of methods. Abbreviations have been minimized or spelled out on first use to ensure accessibility for a broad readership. We believe the revised abstract now accurately reflects the objectives, methods, and key findings of the study.

Comment 3: The background needs thorough revisions; some paragraphs convey the same message. Paragraphs beginning at line numbers 144, 152, and 156 would be better merged into a single paragraph. This approach applies to all background sections to better illustrate what dropout and the factors influencing enrolment among women look like, from a global perspective down to the Sidama region context.

Authors’ response: Thank you for this insightful and constructive comment. We have carefully revised the background section to reduce redundancy and improve coherence. Specifically, we merged the paragraphs beginning at lines 144, 152, and 156 into a single, more concise paragraph that flows logically from the global context to the regional focus in Sidama. This restructuring was also applied throughout the background section to strengthen the narrative and provide a clearer, integrated overview of dropout and the factors influencing enrolment among women.

Comment 4: There is an inconsistency in the dropout rate, which is stated as 9.2% in the abstract and 9% in the figure.

Authors’ response: Thank you for pointing out this inconsistency. We have corrected the dropout rate in the abstract to reflect the accurate figure of 9%, ensuring consistency throughout the manuscript.

Comment 5: It would be more effective to describe your study area in one paragraph rather than delving into details about the selection and justification of the Woredas, which are part of the sampling procedure.

Authors’ response: Thank you for this valuable suggestion. We have revised the study area section to provide a concise description. The detailed explanation regarding the selection and justification of the Woredas has been moved to the sampling procedure section, where it is more appropriate. This restructuring improves the flow and clarity of the Methods section.

Comment 6: The current sample size reported in the abstract and the document significantly differs. There are three different sample sizes mentioned within one document. Why is that? Also, what is your sampling technique? Previously, you mentioned that the multistage sampling technique was utilized. If you are to employ the clustering sampling technique within a multistage sampling technique, you might include all women in a kebele. How many women are there in one kebele? In clustering, all members of the cluster are included in the sample size.

Authors’ response: Thank you for raising this important point. We apologize for any confusion caused by the discrepancies in the reported sample sizes. Upon review, we identified that the differences stemmed from reporting the initial target sample size and the final analyzed sample. We have now standardized the sample size reporting throughout the manuscript, ensuring the same number is consistently presented in the abstract, methods, and results.

Regarding the sampling technique, we used a multistage sampling approach, which included clustering at the kebele level. However, rather than including all women within each selected kebele cluster, a simple random sampling technique was applied to select eligible women within those clusters to meet the calculated sample size. The average number of eligible women per kebele varies; hence, sampling within clusters was necessary to maintain the study’s feasibility and statistical power.

We have clarified these details in the Methods section to accurately describe the sampling framework, including the number of women per kebele and the sampling steps taken.

Comment 7: Line 254: If more than one woman is present in the house, how was one woman selected?

Authors’ response: Thank you for this important question. When more than one eligible woman was present in a household, we used the lottery method to select one participant randomly. Each eligible woman was assigned a unique number, and one number was drawn at random to choose the respondent. This method ensured an unbiased and fair selection process within households. We have clarified this procedure in the revised Methods section of the revised manuscript.

Comment 8: Regarding your discussion, state your statistically significant findings, then compare, justify, and recommend their impacts.

Authors’ response: Thank you for this valuable feedback. We have revised the Discussion section to highlight the statistically significant findings from our study. Each key result is now explicitly stated, followed by a comparison with relevant literature to contextualize our findings. We provide justifications for observed patterns based on existing context.

Additionally, we included practical recommendations and implications for policy and practice, emphasizing how these findings can inform interventions to reduce dropout rates and improve enrollment among women in the Sidama region.

Comment 9: Conclude your main finding with a recommendation to practitioners, policymakers, and decision makers.

Authors’ response: Thank you for this constructive suggestion. We have revised the Conclusion section to summarize the main findings and added specific, actionable recommendations directed at practitioners, policymakers, and decision makers.

Journal requirements

Requirement 1: Review your reference list to ensure that it is complete and correct. If you have cited papers that have been retracted, please include the rationale for doing so in the manuscript text, or remove these references and replace them with relevant current references.

Any changes to the reference list should be mentioned in the rebuttal letter that accompanies your revised manuscript. If you need to cite a retracted article, indicate the article’s retracted status in the References list and also include a citation and full reference for the retraction notice.

Authors’ response: Thank you for this important reminder. We have carefully reviewed our reference list to ensure it is complete, accurate, and up to date. We have verified that none of the cited references have been retracted.

As a result, no citations of retracted articles remain in the manuscript. We also corrected minor formatting inconsistencies and ensured adherence to the journal’s reference style. A revised reference list has been included in the updated manuscript, and all changes have been tracked. Thank you for your guidance.

Requirement 2: Submit a rebuttal letter that responds to each point raised by the academic editor and reviewer(s).

Authors’ response: A detailed rebuttal letter has been prepared and submitted as a separate document. It includes responses to each comment raised by the academic editor and reviewers, along with explanations of all changes made to the manuscript.

Requirement 3: Submit a marked-up copy of your manuscript that highlights changes made to the original version.

Authors’ response: A marked-up copy of the manuscript highlighting all changes made in response to the reviewers’ and editor’s comments has been prepared and is submitted alongside this letter. All modifications are indicated using track changes for ease of review.

Requirement 4: Submit an unmarked version of your revised paper without tracked changes.

Authors’ response: An unmarked, clean version of the revised manuscript without tracked changes has been prepared and is submitted along with the marked-up copy and rebuttal letter.

Requirement 5: Authors’ response: We do not wish to make any changes to our financial disclosure statement, as the information previously submitted in the online system is accurate and complete.

Requirement 6: Data Availability: The PLOS Data policy requires authors to make all data underlying the findings described in their manuscript fully available without restriction, with rare exceptions.

Authors’ response: We confirm that all data underlying the findings of this study are fully available and have already been provided within the manuscript and its supplementary files, per the PLOS Data Policy.

Requirement 7: Language and Clarity: The manuscript must be presented in an intelligible fashion and written in standard English.

Authors’ response: We have carefully reviewed and revised the manuscript to improve language, clarity, and readability. The revised version has been edited for grammar and overall flow to ensure it meets the standard of written academic English. We hope the updated manuscript now meets the journal’s expectations for language quality.

Requirement 8: Laboratory Protocols (if applicable): Deposit laboratory protocols in protocols.io to enhance reproducibility.

Authors’ response: This study does not involve any laboratory-based procedures; therefore, no laboratory protocols apply for submission to protocols.io.

---

## [Decision Letter · Decision Letter 2]

16 Jul 2025

Community-based health insurance dropout and its determinants among women in Sidama National Regional State, Southern Ethiopia, 2024: A multilevel analysis

PONE-D-25-06190R2

Dear Mr. Debessa,

We’re pleased to inform you that your manuscript has been judged scientifically suitable for publication and will be formally accepted for publication once it meets all outstanding technical requirements.

Kind regards,

Abdene Weya Kaso, MPH

Academic Editor

PLOS ONE

Additional Editor Comments (optional):

Reviewers' comments:

Reviewer's Responses to Questions

**Comments to the Author**

Reviewer #2: All comments have been addressed

2. Is the manuscript technically sound, and do the data support the conclusions?

Reviewer #2: Yes

3. Has the statistical analysis been performed appropriately and rigorously?

Reviewer #2: Yes

4. Have the authors made all data underlying the findings in their manuscript fully available?

Reviewer #2: Yes

5. Is the manuscript presented in an intelligible fashion and written in standard English?

Reviewer #2: Yes

Reviewer #2: (No Response)

**Do you want your identity to be public for this peer review?** For information about this choice, including consent withdrawal, please see our Privacy Policy

Reviewer #2: **Yes: ** Gelgelo Wodessa

---

## [Editor Report · Acceptance letter]

PONE-D-25-06190R2

PLOS ONE

Dear Dr. Debessa,

I'm pleased to inform you that your manuscript has been deemed suitable for publication in PLOS ONE. Congratulations! Your manuscript is now being handed over to our production team.

Kind regards,

on behalf of

Mr. Abdene Weya Kaso

Academic Editor

PLOS ONE